# Effects of Unconventional Work and Shift Work on the Human Gut Microbiota and the Potential of Probiotics to Restore Dysbiosis

**DOI:** 10.3390/nu15133070

**Published:** 2023-07-07

**Authors:** Aroa Lopez-Santamarina, Alicia del Carmen Mondragon, Alejandra Cardelle-Cobas, Eva Maria Santos, Jose Julio Porto-Arias, Alberto Cepeda, Jose Manuel Miranda

**Affiliations:** 1Laboratorio de Higiene Inspección y Control de Alimentos, Departamento de Química Analítica, Nutrición y Bromatología, Campus Terra, Universidade de Santiago de Compostela, 27002 Lugo, Spain; aroa.lopez.santamarina@usc.es (A.L.-S.); alicia.mondragon@usc.es (A.d.C.M.); alejandra.cardelle@usc.es (A.C.-C.); josejulio.porto@usc.gal (J.J.P.-A.); josemanuel.miranda@usc.es (J.M.M.); 2Área Académica de Química, Universidad Autónoma del Estado de Hidalgo, Carretera Pachuca-Tulancingo km. 4.5, Pachuca 42076, Hidalgo, Mexico; emsantos@uaeh.edu.mx

**Keywords:** workers, gut microbiota, military, seafarers, healthcare, farmers, probiotics

## Abstract

The work environment is a factor that can significantly influence the composition and functionality of the gut microbiota of workers, in many cases leading to gut dysbiosis that will result in serious health problems. The aim of this paper was to provide a compilation of the different studies that have examined the influence of jobs with unconventional work schedules and environments on the gut microbiota of workers performing such work. As a possible solution, probiotic supplements, via modulation of the gut microbiota, can moderate the effects of sleep disturbance on the immune system, as well as restore the dysbiosis produced. Rotating shift work has been found to be associated with an increase in the risk of various metabolic diseases, such as obesity, metabolic syndrome, and type 2 diabetes. Sleep disturbance or lack of sleep due to night work is also associated with metabolic diseases. In addition, sleep disturbance induces a stress response, both physiologically and psychologically, and disrupts the healthy functioning of the gut microbiota, thus triggering an inflammatory state. Other workers, including military, healthcare, or metallurgy workers, as well as livestock farmers or long-travel seamen, work in environments and schedules that can significantly affect their gut microbiota.

## 1. Introduction

The work pressure and stressful conditions to which workers are exposed in certain jobs can affect their health and safety [1]. Currently, it is estimated that 33% of employees work outside of their country’s regular working hours. In fact, according to the US Bureau of Labor Statistics, there are approximately 15 million Americans who work late afternoons, evenings, rotating shifts, or other irregular hours [2]. In Europe, almost one in four workers work night shifts [3]. These work schedules affect the social lives of workers, causing frequent changes in their schedules and lifestyle habits, including changes in sleep patterns and eating habits. Moreover, these unconventional schedules are associated with increased morbidity among workers [4].

In terms of dietary behavior, it was reported that the total energy intake, either on average or in kcal or kcal/week, of workers on a rotating shift schedule does not differ significantly from those working on a conventional schedule [5]. However, there are significant differences in the distribution of caloric intake in the different meals of the day, as well as in the type of food intake [5]. In fact, shift workers have been found to have lower intakes of dietary fiber (because of lower fruit and vegetable intake). This has been linked to excessive energy intake and the development of overweight and obesity, as foods containing higher amounts of dietary fiber prevent overweight and obesity by avoiding excessive calorie intake through satiety-based signaling [6]. In addition, workers tend to compromise on variety in their diet, often eating on a prearranged schedule that is out of sync with the individual’s circadian rhythms. For example, they substitute individual foods for mixed meals and opt for fast or simple meals that are often characterized by a low nutritional value, high-calorie density, high sodium content, and excessive consumption of sugary drinks [7]. In fact, on average, during the first year of employment, new shift workers tend to increase their body mass index (BMI) by about one point, which is attributable to a combination of factors, such as higher-calorie food choices during the working day, lower diet quality, and reduced time spent on leisure activities [8].

In addition, it is known that working hours, the type of work, or even the job position are factors that can affect the gut microbiota (GM) of workers. These alterations may increase the likelihood of developing various metabolic and inflammatory diseases, such as obesity, diabetes, inflammatory bowel diseases, and asthma [9]. In addition, they may also increase the risk of neurological disorders, such as depression, anxiety, and Parkinson’s disease [10,11]. It has been shown that an altered feeding rhythm can alter the GM, leading to dysbiosis [12]. Recent research has indicated the existence of a “circadian clock-gut microbiota axis” in which the GM regulates the circadian clock regulation, with evidence of bidirectional communication between them [13,14,15,16]. Currently, studies on changes in the GM due to shift work and jet lag focus mainly on the disruption or interruption of this axis [17]. Several recent studies have also shown that GM is crucial for the pathogenesis of stress-related diseases and neurodevelopmental disorders, in addition to the development of brain function [18]. Indeed, it has been shown that even short-term exposure to stress can alter the relative proportions of key human GM phyla [19]. Similarly, experimental modifications of the human genome influence the stress response, anxiety, and hypothalamic-pituitary-adrenal neuroendocrine stress axis [18]. In addition, recent studies suggest that the GM may change depending on the type of work performed, for example, when exposed to chemicals, metals, and particulate matter, and that the GM may modulate the effect of these exposures [12].

The aim of this review is to gather information on how unconventional work affects the health and the GM of workers, as well as the effect of using probiotics to modulate the GM, restore dysbiosis, and improve the quality of life of these workers.

## 2. Effects of Work in Unconventional Schedules on Human Health

Shift work refers to work activities that take place at times other than traditional daytime hours. While it is true that alternative working hours are often necessary and can facilitate productivity, it is also important to recognize that these unconventional working hours can have an adverse impact on our health and well-being [20]. Several observational studies have shown that shift workers face an increased risk of developing obesity, type 2 diabetes mellitus (DM2), and metabolic syndrome compared to those working equivalent daytime hours [3]. In addition, as the years of shift work accumulate, the risk increases. This phenomenon applies to both night work and rotating shifts [3].

Unconventional shift work, especially at night, is associated with an increased risk of cardiovascular disease, cancer, diabetes, hypertension, chronic fatigue, sleep disorders, increased body weight, and therefore, a higher mortality rate [20,21,22]. Workers on rotating shifts have a higher prevalence of irritable bowel syndrome, abdominal pain, constipation, and diarrhea than workers on conventional schedules [23,24,25,26]. It has been suggested that inadequate nutrition, irregular meal timing, and psychological disorders may contribute to this high prevalence [5]. On the other hand, workers with unconventional work schedules are more likely to suffer from anxiety symptoms due to work schedule problems and disturbed sleep patterns, triggering increased anxiety levels [27].

### 2.1. Changes in Dietary Pattern

Most workplaces have shifted to less physically demanding tasks (for example, automation), while portability and the availability of dense, high-energy “snacks” have increased in the workplace. This may also be due to an increase in the intake of foods with excess fats, oils, and refined flour, such as junk food. In addition, a lack of physical activity is triggered by a lack of time and feeling more tired. For these reasons, the need to implement nutrition education programs combined with physical exercise is the greatest among shift workers.

A meta-analysis of 24 studies [28] that targeted these workers and included measures as simple as offering fruit at reduced prices in vending machines, providing access to professional nutritional advice, or encouraging physical exercise by giving away sports passes or organizing sports competitions among company members achieved an average weight reduction of 1.2 kg and a decrease in BMI of 0.47 kg/m^2^ after one year of implementation. At first glance, this decrease may not seem very significant, but considering that most of the population aged 18–49 years gains, on average, between 0.5 and 1 kg per year, it cancels out the average weight gain resulting from the increasing age of the workforce. Shift workers have been found to have a higher BMI, higher cholesterol and triglyceride levels, and a higher risk of high blood pressure than workers on conventional schedules. Therefore, their risk of developing cardiovascular diseases is up to 40% higher than in conventional shift workers [8,29].

In terms of feeding patterns [30], mice exposed to a 4-week schedule were observed to experience changes in their feeding patterns, consuming food irregularly. This altered feeding schedule was significantly associated with a loss of circadian rhythm and changes in the composition of the gut microbiome in mice. In addition, an association was found between an increased F/B ratio and changes in fat mass and inflammation. Based on these findings, it was hypothesized that the changes in the microbiome observed from baseline to the first energy restriction period might be repeated, and possibly more pronounced, in samples collected during the second energy restriction period [30].

### 2.2. Hormonal Changes

The elevated risk of cardiometabolic problems in shift workers may also be due to hormones involved in metabolism, such as ghrelin (which increases appetite) and leptin (which reduces appetite), which are regulated by transcription factors in the biological clock. These hormones play a key role in controlling appetite and eating behavior by sending stimulatory and inhibitory signals to the central nervous system, in particular the hypothalamus [31]. During the night, especially in the time interval between 9 p.m. and 4 a.m., circulating blood levels of ghrelin remain very high and are accompanied by low circulating levels of leptin [31]. This imbalance in the ghrelin/leptin ratio means that people who are awake at this time of the day have a greater appetite and are therefore more likely to gain weight. Moreover, this imbalance is intensified in people who suffer from sleep deprivation and/or poor sleep quality. Therefore, this imbalance will chronically increase in workers on rotating or night shifts, which partly explains their higher propensity to obesity [31]. Shift work can also lead to increased circulating levels of resistin, an endocrine hormone implicated in the development and progression of insulin resistance, as well as cortisol, a hormone traditionally associated with stress. This is probably related to a lower insulin response at night, which is an important activator of lipoprotein lipase, an enzyme found in the capillary walls that stimulates the hydrolysis and removal of triacylglycerol [31]. A large body of scientific literature published in recent years demonstrate a link between shift and/or night work and insulin resistance, diabetes, dyslipidemia, and metabolic syndrome [32]. The link between shift work and higher insulin resistance is thought to be due to decreased melatonin production. Melatonin is a key factor in insulin synthesis, secretion, and action. In addition, its concentration also regulates the expression of the glucose transporter GLUT 4. Relevant work in this area has shown that shift work leads to increased insulin resistance and an increased inflammatory state [32]. In addition, cortisol levels have been shown to increase in stressful situations and during prolonged low-calorie diets. These two factors are closely related to shift work, so when stress or blood pressure increases, cortisol levels also increase. When cortisol is high, it is very difficult for the body to regulate blood sugar, and it also inhibits the loss of body fat and the gain in muscle mass. The same is true when the duration of sleep is short and cortisol levels rise and exacerbate the above [33].

Currently, chronic low-grade inflammation has been recognized to play an important role as a biological factor in stress-related ill health [34]. Both sleep disturbance and sleep deprivation have been reported to be associated with changes in different immunological parameters that favor a pro-inflammatory state, which may underlie the negative health effects of sleep disturbance. The slightest loss of sleep has negative health consequences, both physiological and mental, as well as negative effects on performance [35]. This evidence suggests that sleep-mediated immune disruption, together with evidence that pathogen-induced immune responses can negatively affect the sleep cycle, demonstrates that there is a bidirectional relationship between sleep and immunity, which may have profound implications in immune-related human diseases [36]. Any disturbance in sleep can affect the physiological systems that govern immune cell distribution and cytokine activity. A normal sleep cycle plays a key role in regulating the hypothalamic-pituitary-adrenal axis, which controls the circadian rhythm. Systemic cytokine levels show an increase in the mid-morning and late afternoon, reflecting glucocorticoid secretion [37].

### 2.3. Sleep Disruption

Sleep loss and acute circadian disruption, whether due to total sleep deprivation, sleep restriction, or sleep fragmentation, have been found to be physiological stressors. These changes are analyzed in relation to fluctuations in cortisol levels that occur with sleep loss and circadian disruption. Cortisol elevation following sleep loss and acute circadian disruption is thought to reflect activation of the hypothalamic-pituitary-adrenal (HPA) axis [38]. Indeed, cortisol has significant effects on glucose metabolism. It affects the body’s glucose balance by stimulating gluconeogenesis (the production of glucose from sources other than glucose) and reducing peripheral glucose utilization, which can lead to an increase in blood glucose levels. In addition, studies investigating sleep duration disruption have found that sleep loss and acute circadian disruption are associated with increased cortisol levels, especially in the afternoon and early evening [38,39]. This suggests that sleep loss may contribute to metabolic dysfunction, at least in part, via the activation of a cortisol-mediated physiological stress response.

Experimental sleep studies suggest that sleep deprivation can increase adrenal gland activity [38]. Consequently, sleep deprivation may generate physiological stress that alters metabolic homeostasis, leading to a series of low-intensity inflammatory processes that contribute to metabolic dysfunction. If true, this could also provide a new theory for the observed relationship between sleep deprivation, circadian dysregulation, and inflammatory markers [40].

It has been hypothesized that sleep restriction may trigger pro-inflammatory changes in the gut microbiome. These changes could drive the metabolic and cognitive effects observed after sleep restriction [41,42,43]. In a randomized study of healthy Caucasian volunteers who followed regular eating and exercise schedules, participants were subjected to two nights of partial sleep deprivation and two nights of normal sleep. It was observed that in fecal samples taken after two nights of partial sleep deprivation, there was an increase in the abundance of Firmicutes bacteria and a decrease in Bacteroidetes bacteria compared to those taken after two nights of normal sleep [41,42]. These changes in relative bacterial abundance are similar to the patterns observed in fecal samples from obese people, as described in a study by Benedict et al. [44]. Human research has shown that the presence of higher counts of Firmicutes, which are more prevalent in rotating and night shift workers [43], is associated with an increased predisposition to obesity and metabolic diseases [43,44]. Another study [24] reinforces the results obtained above by showing that humans who experienced an 8 h circadian offset due to flights across different time zones showed a change in the composition of the gut microbiome and a pronounced increase in the phylum Firmicutes [24].

Several previous studies, such as that by Zhang et al. [43], have shown that in sleep-restricted rats, an increased Firmicutes/Bacteroidetes (F/B) ratio, which has been associated with changes in fat mass and inflammation, is observed during rest. There is support for the idea that cortisol levels in the afternoon or early evening may be altered by shorter sleep periods or lack of sleep, as well as by acute circadian mismatch. This physiological response to stress triggered by sleep deprivation is likely to be related to changes in GM. The above-mentioned detailed relationship between cortisol and the GM suggests a possible explanation for the connection between stress triggered by sleep disruption or sleep deprivation, circadian disruption, and possible changes in the GM [43].

### 2.4. Work Environment

As mentioned above, the community composition and diversity of GM microorganisms are susceptible to change and are affected by various factors, such as diet, activity, sleep patterns, genetics, drugs, and environment, including the type of work and shift work (Figure 1) [45,46].

Sleep loss and acute circadian disruption are considered physiological stressors that contribute to this susceptibility [46]. For example, in a study by Thaiss et al. [24], jet lag was induced in a group of mice by subjecting them to an 8 h advance in their daily schedule for three days and showed evidence that host circadian misalignment results in microbial dysbiosis, leading to metabolic imbalances.

On the other hand, it is necessary to consider that a specific work environment can also influence the GM and thus the health of workers [47,48]. Table 1 lists the different studies related to the environment of different types of unconventional work and their effects on GM. For example, the chemical and physical conditions present in hospitals give rise to microbial communities that differ widely from those found in the natural external environment [49]. Even within the same workplace, different areas often have their own microecological characteristics. For example, in hospital intensive care units (ICUs), where intensive medical activities are conducted, and frequent exposure to infectious pathogens occurs, microbial communities are particularly hazardous [49,50,51]. Indeed, the microbiology of the hospital environment has been shown to impact the health of hospital staff [52]. In addition, healthcare workers have been found to have a higher incidence of *Clostridium difficile* infections in their workplaces, suggesting that they may act as potential vectors of infectious diseases [53,54].

There is a bidirectional relationship between the GM and circadian rhythm. Thus, the circadian clock influences the composition of the GM; similarly, the GM also regulates the circadian rhythm. Studies are currently focusing on the disruption of this bidirectional axis [12]. Indeed, human GM has been shown to play a key role in the development of brain function, and consequently, in stress-related diseases and neurodevelopmental disorders [12].

## 3. Effects of Different Types of Unconventional Work on the Human Gut Microbiota

A link between impaired GM and type II diabetes mellitus (DM2), as well as metabolic syndrome, has also been demonstrated [59]. The physiological and psychological stress that occurs during shift work negatively influences intestinal permeability and affects health via impaired GM. In a healthy gut environment, the epithelial barrier is a well-maintained structure designed to restrict the impact of pathogens and promote and support the fight against ‘beneficial’ inflammatory bacteria [59]. Maintaining an appropriate balance between beneficial and pathogenic bacteria in the mucosa and within the gut is crucial for gut stability. Both the integrity of the epithelial barrier and the protective gut environment can be disrupted by various environmental challenges, including stress [59].

In the human GM, it has been observed that approximately 10–35% of the operational taxonomic units (OTU) show diurnal variations in abundance, according to Bijnens and Depoortere [3]. For example, oscillations in the abundance of *Parabacteroides*, *Lachnospira,* and *Bulleida* have been identified, as mentioned in a study by Thaiss et al. [24]. In addition, short-chain fatty acids (SCFA) show a rhythmic pattern and decrease throughout the day in humans [3]. In vivo studies have shown that oral administration of SCFA induces a phase shift in several peripheral tissues, as observed in a study by Tahara et al. [60]. Importantly, the change in microbiome composition observed in jet-lagged individuals was completely restored after two weeks of recovery from jet lag [24].

### 3.1. Healthcare Workers

A study by Zheng et al. [57] showed marked differences in the diversity and structure of the GM between healthcare workers and non-healthcare workers. Short-term contract workers were found to have higher microbial diversity than long-term contract workers. Firmicutes, Bacteroidetes, Proteobacteria, and Actinobacteria were found to constitute the majority of the predominant phyla in the GM of all participants. However, the phylum Firmicutes was found to be more abundant in short- and long-term workers than in non-medical individuals. On the other hand, the phylum Bacteroidetes were less abundant among healthcare workers. As for Proteobacteria, they were slightly more abundant in short-term workers, although this difference was not statistically significant [57].

Alterations in the microbiomes of medical workers may have important implications for their health. In this study [57], a deviation in the distribution of medical workers’ enterotypes was observed, with an increase in Firmicutes abundance and a decrease in Bacteroides and *Prevotella* levels. It was assessed whether the hospital department (ICU vs. non-ICU) and job position (resident physician vs. nursing staff) of the medical workers had an impact on the composition of the GM. Significant differences in GM composition between individuals belonging to different hospital departments and occupying different job positions were observed in both short- and long-term groups of workers [57]. Significant differences in the composition of the microbiome were observed between ICU and non-ICU workers. Compared to non-ICU workers, it was observed that ICU workers showed a significant increase in the abundance of *Dialister* bacteria, *Enterobacteriaceae*, *Phascolarctobacterium*, *Pseudomonas*, *Veillonella*, and *Streptococcus*, and a marked decrease in *Faecalibacterium*, *Blautia*, and *Coprococcus* bacteria.

It was observed that the GM of ICU workers, who specialize in the treatment of critically ill patients, shows a significant increase in *Enterobacteriaceae*, Pseudomonas, *Veillonella,* and *Streptococcus* bacteria [57]. These findings support the idea that exposure to the hospital environment and interaction with critically ill patients may influence the composition of the microbiome of ICU workers, particularly the abundance of certain bacterial genera. Members of the *Enterobacteriaceae* family are common ICU pathogens. These bacteria are among the pathogens most frequently associated with nosocomial infections in hospital settings, including ICUs [57]. The enrichment of Enterobacteriaceae members in medical workers could be related to exposure to the ICU environment [57].

The risk of colonization by *C. difficile*, a bacterium associated with nosocomial infections, tends to increase steadily during hospitalization. This suggests that there is a cumulative risk of daily exposure to *C. difficile*, possibly in the form of spores, which may persist in the hospital environment for months [53]. These spores could be a potential source of infection and contribute to the transmission of *C. difficile* to the microbiome of exposed medical workers. Colonization with non-toxigenic strains of *C. difficile* may offer some protection against the development of disease caused by toxigenic strains. This means that the presence of non-toxigenic strains in the microbiome may compete with toxigenic strains, thus limiting their growth and pathogenic activity [53].

### 3.2. Farm Workers

Another worker group studied was farmers. Studies have been conducted on the impact on the microbiota of people working on farms with animals compared to those working in urban environments [55]. Farmers had higher microbial richness and diversity compared to individuals living in urban environments. These results support the hypothesis that residing in urban environments may imply exposure to a less diverse microbial flora, which is associated with an increase in allergic and inflammatory diseases. In addition, they showed a greater similarity to the gut profile of pigs, as evidenced by an increase in bacteria of the genus *Bacteroides* and the family *Clostridiaceae*, as well as a decrease in bacteria of the phylum Firmicutes [55].

Only one study has investigated changes in the oral microbiota associated with exposure to agricultural pesticides. Stanaway et al. [61] found a persistent association across seasons between the detected blood concentration of the insecticide azinphos-methyl and the taxonomic composition of the oral microbiome, specifically a significant reduction in the genus *Streptococcus* [62].

### 3.3. Military Personnel

Another group in which microbiota-related studies have been conducted is the military personnel. In the study by Walters et al. [63], the relationship between variations in the microbiota of soldiers overseas and the incidence of traveler’s diarrhea (TD) infection was investigated. *Ruminococcaceae* was found to be more abundant in soldiers who tested positive for TD, while *Ruminiclostridium* spp. had a higher relative abundance in soldiers who tested negative for TD. In addition, *Haemophilus* spp. and *Turicibacter* spp. were found to be associated with the alleviation of gastrointestinal distress [63].

### 3.4. Long-Travel Seamen

On the other hand, seafarers also face extreme situations and disruption of conventional patterns [64]. Indeed, the difficult conditions they experience during long sea voyages increase the risk of illness and death compared to workers on land. The ocean environment presents a variety of adverse conditions, such as high humidity, high salinity, intense exposure to sunlight and ultraviolet radiation, heavy waves, monotonous environments, disrupted circadian rhythms, sleep disturbances, and limited access to fresh fruits and vegetables [64,65]. In addition to vitamin deficiency, chronic diseases affecting the immune system and digestive tract have become the main health risk for people working at sea [62]. The human GM plays a key role in the host immune system and is essential for maintaining human health [66]. Recent studies have highlighted the importance of the diversity of gut microbial species and functional genes in the development of various chronic metabolic diseases. Although attention has been paid to the health of seafarers over long voyages, research in this area remains limited [62,67]. In a study by Zhang et al. [68], it was shown that a long sea voyage not only led to changes in the composition of the GM but also reduced the diversity of its functional characteristics. In this study, the microbiome of sailors in the placebo group underwent significant alterations during a long sea voyage, involving changes in key gut bacterial species and a significant reduction in genes for carbohydrate-active enzymes, which are critical for maintaining GM homeostasis and host health [69]. These enzymes help us break down these complex carbohydrates into SCFA, either directly or via a cross-feeding mechanism [70]. This, in turn, promotes gut health and improves fitness. Conversely, the low diversity of gut microbes possessing functional genes represented by these degradative enzymes is closely associated with the growth of pathogens that can trigger chronic diseases [68]. At the same time, during prolonged sea voyages, the limited availability of fresh fruits and vegetables may also be a crucial factor contributing to a decrease in gut microbial diversity [67].

### 3.5. Metal and Tunnel-Workers

GM and health are also affected in workers exposed daily to dust, metalworking fluids (MWFs), and pesticides. Two studies investigated possible changes in the GM in workers exposed to silica and ceramic dust. A study by Zhou et al. [71] analyzed the characteristics of the GM in patients with early stage pulmonary fibrosis caused by daily exposure to silica in the work environment. It was observed that, at the phylum level, the abundance of Firmicutes and Actinobacteria was lower in patients with silicosis than in healthy control individuals. These findings may be useful for the early diagnosis of silicosis and prevention of pulmonary fibrosis [71]. Furthermore, Ahmed et al. [72] conducted a study on the composition of the nasal microbiota in workers exposed to dust in ceramic factories. It was observed that dust-exposed workers had a significant increase in the relative abundance of the phylum Proteobacteria, specifically *Haemophilus* spp., with a lower presence of Actinobacteria and Bacteroidetes compared to controls [72].

Another study by Wu et al. [73] performed an invasive characterization of the microbiota in lung biopsies obtained from workers exposed to MWFs. Lung biopsies were performed on symptomatic MWF-exposed workers and showed the presence of bacterial species characteristic of MWFs in a new and distinctive MWF-related lung condition. This condition was characterized by lymphocytic bronchiolitis and alveolar ductitis, with B-cell follicle formation and emphysema [74]. Wu et al. [73] also showed the presence of an OTU designated as Pseudomonas in the lung, skin, and nasal samples from exposed workers, as well as in MWFs. Traces of *Pseudomonas pseudoalcaligenes* were found in these samples, and these readings were a close match for the *P. pseudoalcaligenes* readings of the metal working fluid samples, which suggests that the bacterial DNA found in the tissue samples could have originated from the metal working fluid. Interestingly, this OTU was not found to be differentially enriched in the air samples, suggesting that the main mode of transmission of this microorganism is via direct rather than airborne contact [73].

In addition, it is interesting to note that the alterations observed in the GM of the tunnel workers were similar to those found in patients with mental disorders, especially mood disorders [1], and point to the importance of maintaining a proper balance in the GM to promote mental and emotional well-being [1]. Regarding the high abundance of Actinobacteria and *Coriobacteriaceae* in patients with mental disorders, as mentioned in the study by Lu et al. [1], a possible correlation with dyslipidemia has been suggested. This suggests that the composition of the GM may be related to metabolic and cardiovascular health in patients with mental disorders. In addition, the presence of Bifidobacterium bacteria has been found to be significantly higher in patients with major depressive disorder and bipolar disorder with current major depressive episodes. Several studies have found lower levels of *Faecalibacterium* in patients with mental disorders, including bipolar disorder [1]. *Faecalibacterium* is a bacterial genus associated with the production of SCFA, which are important metabolites for gut health. A decrease in SCFA-related bacteria, such as *Faecalibacterium*, could indicate a dysbiosis in anti-inflammatory activities in workers and possibly in patients with mental disorders. SCFA, produced by intestinal bacteria during the fermentation of food substrates, can modulate the immune response and have anti-inflammatory effects. Therefore, a decrease in these bacteria and the corresponding SCFAs could be related to an inflammatory state and dysfunction in microbiota-gut-brain interactions. Furthermore, it has been postulated that SCFA may directly or indirectly mediate GM-brain interactions via various signaling pathways, such as immune, endocrine, neural, and humoral pathways [1].

## 4. Use of Probiotics to Restore Dysbiosis in Workers

Due to studies showing that both the type of work and unconventional working hours can negatively affect the GM, the use of probiotics has been suggested to alleviate these consequences. Table 2 compiles studies showing the effects of probiotic supplementation on GM and biochemical markers in workers. In fact, they could be used in a personalized way for everyone under the concept of precision probiotics [75].

Probiotics are defined as live microorganisms that, when consumed in adequate amounts, provide health benefits to the host [76]. Commonly, probiotics include bacterial strains of the genera *Lactobacillus* and *Bifidobacterium* [77,78]. To date, most probiotic-related studies have focused on describing variations in the GM and have concluded that limited structural changes in the GM are a common phenomenon following probiotic consumption. These changes are characterized by an increase in the number of beneficial microorganisms and a decrease in pathogens [79,80]. Some clinical studies have investigated the application of probiotics in the treatment of various diseases. For example, their efficacy has been studied in liver disease [81], cardiovascular disease [82], kidney disease [83], irritable bowel syndrome [84], relief of allergy symptoms [85], and against viral infections [82]. Although the results of all these studies were not necessarily positive, probiotics have been found to have common effects on the regulation of GM.

**Table 2 nutrients-15-03070-t002:** Effect of probiotic supplementation of gut microbiota and biochemical markers of workers.

Type of trial	Subjects	Dosage and Time of Exposition	Effects on Gut Microbiota	Other Health Effects	Reference
Double-blind parallel-group trial	94 shift workers	*Lactobacillus acidophilus*, *Bifidobacterium animalis* spp. *lactis* or placebo; 1 × 10^10^ colony count units (cfu) for 14 days	Not investigated	Probiotic supplementation decreased serum markers such as cortisol, pentraxin, or interleukin-1ra, related to sleep quality.	West et al. [22]
Randomized controlled trial	961 women healthcare workers	1.12 × 10^9^ cfu or more of *Lactobacillus bulgaricus* OLL1073R-1 and strain of *Streptococcus thermophilus* daily for 16 weeks	Not investigated	A significant increase in interferon-γ production was found with a daily intake of OLL1073R-1 yogurt.	Kinoshita et al. [86]
A randomized, double-blind, placebo-controlled trial	70 petrochemical workers	100 g/day probiotic yogurt which contained two strains of *Lactobacillus acidophilus* LA5 and *Bifidobacterium lactis* BB12 with a total of min 1 × 10^7^ cfu for 6 weeks	Not investigated	The consumption of probiotic yogurt had beneficial effects on mental health parameters in petrochemical workers.	Mohammadi et al. [87]
A double-blind, randomized, placebo-controlled experiment	41 female healthcare workers employed on a rotating shift schedule	4 g/day of freeze-dried powder of the multistrain probiotic mixture (2.5 × 10^9^ cfu/g) for 6 weeks. The prebiotic mixture contained *Bifidobacterium bifidum* W23, *Bifidobacterium lactis* W51, *Bifidobacterium lactis* W52, *Lactobacillus acidophilus* W37, *Lactobacillus brevis* W63, *Lactobacillus casei* W56, *Lactobacillus salivarius* W24, and *Lactococcus lactis* (W19 and W58)	Not investigated	Results indicate a potential protective effect of probiotics against fat mass gain. Probiotics may alleviate anxiety and fatigue in shift-working females.	Smith-Ryan et al. [88]
A 30-day longitudinal experiment	82 sailors during a sea voyage	2 g package containing mixed probiotics including 9.70 Log cfu of *Lactobacillus casei*, 9.70 Log cfu of *Lactobacillus plantarum* P-8, 9.70 Log^10^ cfu of *Lactobacillus rhamnosus* M9, 9.88 Log^10^ cfu of *Bifidobacterium lactis* V9, and 9.88 Log^10^ cfu of *Bifidobacterium lactis* M8 once daily for 30 days	The compositions of the intestinal microbiota of the two groups (placebo and probiotic) were highly distinct at the end of the sea voyage, which confirmed the positive impacts of the probiotics consumed.	Probiotics maintained intestinal microbiome homeostasis and further prevented anxiety during the long sea voyage.	Zhang et al. [89]
A randomized, double-blind placebo-controlled study	262 employees (day workers and three shift workers)	Daily dose of 10^8^ cfu of *L. reuteri* for 80 days	Not investigated	Among the 53 shift workers, 33% in the placebo group reported being sick during the study period compared to none in the *L. reuteri* group.	Tubelius et al. [90]
Open-label single-arm study	90 highly stressed information technology specialists	300 mg of lyophilized *L. plantarum* PS128^TM^ powder, which is equivalent to 10 billion cfu for 8 weeks	Not investigated	Significant improvements in self-perceived stress, overall job stress, job burden, cortisol level, general or psychological health, anxiety, depression, sleep disturbances, quality of life, and both positive and negative emotions	Wu et al. [73]
A controlled, parallel, randomized, and double-blind clinical trial	65 military	Symbiotic ice cream containing: 2 × 10^8^ cfu/g for *L. acidophilus* LA-5 and 2.7 × 10^9^ cfu/g for *B. animalis* BB-12 and 2.3 g of inulin in the 60 g of ice cream for 30 days	No significant differences in α diversity between groups. No significant differences in the proportions of each phyla comparing the two groups.	This supplementation improved tenseness and sleepiness in healthy young military.	Valle et al. [91]

Evidence of the beneficial effects and immune-modulating capacity of probiotics suggests that investigating the use of probiotic supplements to ameliorate sleep disruption-induced changes in night work-associated inflammation is a promising avenue [92]. It is certain that probiotics can transiently colonize the GM [92,93]. It has been postulated that these microorganisms exert beneficial health effects by modifying the immune system.

A study by West et al. [22] investigated the acute effects of two different probiotic strains, *Lactobacillus acidophilus* DDS-1 and *Bifidobacterium animalis* spp. *lactis* UABla-12, on the immune system of night shift workers. These strains, alone or in combination, have previously been shown to reduce the severity of abdominal pain and irritable bowel syndrome symptoms [88,90,93], improve functional constipation [94], provide symptomatic relief in cases of lactose intolerance [94], and show protective effects against atopic dermatitis [95] and respiratory infections [96]. Night shifts are associated with significant changes in the markers of stress and immunity, and probiotic supplementation was found to moderate the severity of these changes. These results suggest that there are wider health implications associated with night work than with circadian disruption [21,97]. They also provide initial evidence for the use of probiotics to attenuate the effects of stress associated with night work.

Another study [27] investigated the acute effects of probiotics on stress indices, acute-phase responses, and inflammation during two nights of night work. It was observed that the most significant changes occurred during the anticipatory stress before the night shift [27]. Anticipatory stress refers to stress associated with upcoming events [90] and is related to illness [98] and dysregulation of the HPA axis and the immune system [99]. The study results provide initial evidence that *L. acidophilus* and *Bifidobacterium lactis* can improve the biological impact of stress and that *B. lactis* can also improve sleep quality. In addition, a high recovery rate of the ingested probiotic species was observed. This study presents a novel approach to understanding the role of probiotic supplementation in inducing stress and immune system alterations. Furthermore, it confirms previous findings that *L. acidophilus* and *B. lactis* can colonize the gut and be detected in feces [94].

On the other hand, there are also studies exploring the possibility that probiotics may play a role in maintaining microbiome homeostasis during long sea voyages. One example is the study by Zhang et al. [68], who investigated the possible effects of probiotics on improving the physical fitness of seafarers. In this study, a questionnaire was designed that included scores related to bowel movement (such as stool consistency and volume; the presence of constipation, diarrhea, and bloody stools; and the frequency of defecation), pain in the stomach, pain in the legs, headache, chest pain, muscle pain, and stress/anxiety at the end of the journey. Following probiotic supplementation, significant differences were found in the reduction in stress and anxiety scores [87,91]. These results indicate that probiotics have the potential to prevent the onset of anxiety during a long sea voyage [72]. Another 30-day longitudinal study on Chinese seafarers revealed that probiotics were able to maintain the homeostasis of the GM during a long sea voyage. The probiotics used in the study included strains such as *Lactobacillus casei*, *Lactobacillus plantarum*, *L. rhamnosus*, and *B. lactis*, which demonstrated excellent probiotic properties in previous studies [68,100].

Another option to reduce anxiety or depression would also be the supplementation of SCFA or dietary fiber intake [41], as recent research indicates that SCFA or the microbiota may influence brain function, such as anxiety or depression [100]. Therefore, it is possible that microbiota-derived metabolites reach the brain, and future studies are required to understand the impact of chronic SCFA administration on the circadian clock system in the brain [41]. The results indicate that bacterial fermentation end products, such as SCFA, show a decrease throughout the day [101]. The two most abundant genera associated with time were *Roseburia* and *Ruminoccocus*, which accounted for 2% and 3% of the bacterial community in our participants, respectively. In addition, both *Roseburia* and *Eubacterium* decreased throughout the day. Both genera are butyrate producers, and a decrease in butyrate concentrations throughout the day was reported [101].

Associations between the time of day and bacterial abundance could be related to specific bacterial characteristics, such as bile resistance. For example, *Oscillospora* and *Bilophila*, which increased during the day, are bile tolerant, which could give them a competitive advantage during waking hours when increased bile secretion occurs due to food intake [101]. Alternatively, these fluctuations could be influenced by factors independent of the presence of food in the gastrointestinal tract. Circadian variations in the murine microbiota are observed even when parenteral nutrition is the only source of food. Other factors may include hormonal signals from the hosts. For example, *Enterobacter aerogenes* may be affected by melatonin, a circadian hormone [101].

A narrative review suggested that probiotics, prebiotics, and postbiotics may improve sleep quality and reduce stress by influencing sleep latency, sleep duration, and cortisol levels. However, it is important to note that stronger evidence and further validation are required to support these findings, as well as to understand the underlying mechanisms in detail. This can be achieved via appropriate methodological adjustments and further research [102].

## 5. Conclusions

The effects of shift and night work, as well as unconventional work, on human health have various aspects related to both personal characteristics and working and living conditions. Shift work is known to be a risk factor for many health disorders, such as gastrointestinal, psychological, and cardiovascular disorders, but it also disturbs the homeostasis of the sleep/wake cycle and circadian rhythms, as well as hinders family and social life, which triggers other problems. Furthermore, there is a plausible basis for the hypothesis that the suppression of melatonin secretion due to shift or night work may contribute to GM dysbiosis. Furthermore, the work environment or working environment may influence GM, leading to gut dysbiosis in many cases. It has been suggested that the use of probiotics could be a mechanism to restore the intestinal microbiota in workers. The results obtained suggest that probiotic supplementation may be effective in protecting and preserving the diversity and stability of the intestinal microbiota under the special conditions present in various jobs. However, more research is still needed on how the microbiota is affected by probiotics and their use at the individual and personalized level.

## Figures and Tables

**Figure 1 nutrients-15-03070-f001:**
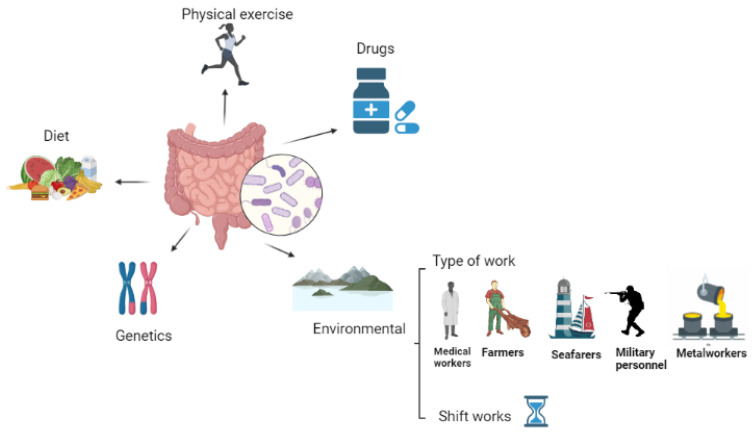
The different factors that can affect the human gut microbiota. Source: own elaboration. (Created in BioRender.com).

**Table 1 nutrients-15-03070-t001:** Effects of work environment on workers’ gut microbiota.

Workers	Subjects	Dosage and Time of Exposition	Effects of Gut Microbiota	Other Health Effects	Reference
**Tunnel miners**	48 healthy men	Before and after 3 weeks of working in a tunnel 8 h/day	Decreased GM diversity; increased Actinobacteria and Bifidobacteriales, Corynebacteriales and Desulfovibrionales; increased *Bifidobacterium*, *Romboutsia*, *Clostridium,* and *Leucobacter*; decreasing *Faecalibacterium* and *Roseburia*	Decreased antioxidant efficacy, digestive and absorptive capacity; increased proinflammatory factors	Lu et al. [1]
**Healthcare workers**	214 workers	Fecal samples were subjected to an enzyme immunoassay for toxins A and b and for glutamate dehydrogenase	Only 0.8% of healthcare workers were found positive for *C. difficile* toxins and antigen	The results found did not confirm the hypothesis that a long stay in the hospital is a risk of *C. difficile* infection	Friedman et al. [53]
**Farm workers**	6 swine farm workers and 6 local villagers	16S rRNA gene sequencing of fecal samples	Workers had less species diversity compared to the local villagers, as well as higher amounts of Proteobacteria and Clostridiaceae	Analysis of antimicrobial resistance genes did not reveal significant differences among workers and villagers	Sun et al. [48]
**Veterinary students in swine farms**	14 students who stayed 3 months on swine farms	91 fecal samples investigated by 16S rRNA and whole metagenome shotgun	Moderate decrease in Bacteroidetes and an increase in Proteobacteria phyla	Antibiotic resistance genes were found in similar genetic contexts in student samples and farm environmental samples	Sun et al. [55]
**Healthcare workers**	71 frontline healthcare workers fighting against COVID-19	A longitudinal investigation at four time points: immediately after they finished treatment and left the isolation wards (Day 0), after a two-week quarantine in a hotel (Day 14), four weeks after their return to normal life (Day 45), and a half year after the frontline work (Day 180)	Microbes associated with mental health were mainly *Faecalibacterium* spp. and *Eubacterium eligens* group spp. Of note, the prediction model indicated that a low abundance of *Eubacterium hallii* group uncultured bacterium and a high abundance of *Bacteroides eggerthii* immediately after the two-month frontline work were significant determinants of the reappearance of post-traumatic stress symptoms.	Stressful events induced significant depression, anxiety, and stress	Gao et al. [56]
**Medical workers**	175 healthy medical workers	Short-term (1–3 months) workers (*n* = 80) and long-term (>1 year) workers (*n* = 95)	Short-term workers: significantly higher abundances of *Lactobacillus*, *Butyrivibrio*, *Clostridiaceae*, *Clostridium*, *Ruminococcus*, *Dialister*, *Bifidobacterium*, *Odoribacter*, and *Desulfovibrio*, and lower abundances of *Bacteroides* and *Blautia*. Long-term workers: higher abundances of taxa such as *Dialister*, *Veillonella*, *Clostridiaceae*, *Clostridium*, *Bilophila*, *Desulfovibrio*, Pseudomonas, and *Akkermansia*, and lower abundances of *Bacteroides* and *Coprococcus*	Not investigated	Zheng et al. [57]
**Nurses**	51 full-time staff nurses	Worked 12 h day or night shifts	No differences in the richness and diversity of species in samples from nurses working day and night shifts	Not investigated	Rogers et al. [2]
**Security officers**	10 male security officers working rotational day/night shifts	After working the day shift (7:00 h–15: 00 h) for 4 weeks and after working the night shift (23:00 h–7:00 h) for 2 weeks. One-off day per week	The phylum with the highest abundance in the day shift was Firmicutes, followed by Bacteroidetes. In the night shift, the relative abundance of Bacteroidetes decreased; however, Actinobacteria and Firmicutes increased. *Faecalibacterium* was found to be a biomarker of day-shift work.	Not investigated	Mortaᶊ et al. [58]

## Data Availability

Not applicable.

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
