# Peer review of "Effects of Unconventional Work and Shift Work on the Human Gut Microbiota and the Potential of Probiotics to Restore Dysbiosis"

_nutrients, 2023, doi:10.3390/nu15133070_

Round 1

Reviewer 1 Report

I have finished my review on your paper. It is a topic of interest to the researchers in the related areas. However, some issues need to be addressed before that. Please refer to my comments and suggestions given in the following:

1.     The abstract should present a comprehensive analysis of the findings derived from your investigation, aiming to highlight the significant results obtained rather than dwelling extensively on the background.

2.     Use uniform words. For example, unconventional work and non-conventional work.

3.     It is crucial to maintain uniformity in formatting throughout the manuscript, including elements such as the capitalization of the first letter in tables and the consistent formatting of author citations. Consequently, the manuscript should be carefully reviewed to ensure that these formatting elements are consistently applied and in accordance with the journal's guidelines.

4.     It is advisable to enhance the logical structure of the manuscript by further classifying the headings based on the content they represent. By refining the headings and aligning them with the corresponding content sections, which will provide a more cohesive framework for understanding the research findings and arguments presented.

5.     In relation to the effects and mechanisms discussed in the article, it is recommended, whenever feasible, to incorporate graphical representations to enhance their clarity and comprehension.

6.     It is recommended to augment the evidence supporting the section 4. Use of probiotics to restore dysbiosis in workers by providing additional quantitative data. By presenting numerical data, the article can provide a more comprehensive understanding of the extent and significance of the observed influence. And, incorporating additional quantitative data would significantly contribute to the comprehensiveness and credibility of the study's findings.

Author Response

Reviewer 1:

I have finished my review on your paper. It is a topic of interest to the researchers in the related areas. However, some issues need to be addressed before that.

We wish to thank the reviewers for his revision of the manuscript. We expect our answers to be the adequate to take the paper into consideration and that it can be published in Nutrients.

Please refer to my comments and suggestions given in the following:

  1. The abstract should present a comprehensive analysis of the findings derived from your investigation, aiming to highlight the significant results obtained rather than dwelling extensively on the background.

Thank you for your comment, the abstract has been modified based on your recommendations, re-organizing the content and citing the most significant reults

  1. Use uniform words. For example, unconventional work and non-conventional work.

Thank you for your comment. In the revised version of the manuscript, the term was unified to "unconventional".

  1. It is crucial to maintain uniformity in formatting throughout the manuscript, including elements such as the capitalization of the first letter in tables and the consistent formatting of author citations. Consequently, the manuscript should be carefully reviewed to ensure that these formatting elements are consistently applied and in accordance with the journal's guidelines.

Thank you for your comments, the proposed modifications have been made. Modifications were performed in Table 1 according to the journal's guidelines.

  1. It is advisable to enhance the logical structure of the manuscript by further classifying the headings based on the content they represent. By refining the headings and aligning them with the corresponding content sections, which will provide a more cohesive framework for understanding the research findings and arguments presented.

Thank you for your comments. According to the suggestions from the Reviewer, the main text was reorganized (maintaining the original content without major modifications), and subheadings were included. In the heading 2. Effects of work in unconventional schedules on human health, it were included as subheadings “changes in dietary pattern”; “hormonal changes”; sleep disruption”; and “work environment”. In the heading 3. Effects of different types of unconventional work on the human gut microbiota. It were included as subheadings “healthcare workers”; “farm workers”; “military personnel”; “long-travel seamen” and “metal and tunnel-workers”.

  1. In relation to the effects and mechanisms discussed in the article, it is recommended, whenever feasible, to incorporate graphical representations to enhance their clarity and comprehension.

Thank you for your comments. A new figure (Figure 1), page 5, has been included in the revised version of the manuscript.

  1. It is recommended to augment the evidence supporting the section 4. Use of probiotics to restore dysbiosis in workers by providing additional quantitative data. By presenting numerical data, the article can provide a more comprehensive understanding of the extent and significance of the observed influence. And, incorporating additional quantitative data would significantly contribute to the comprehensiveness and credibility of the study's findings.

Thank you for your comments. While we would like to be able to provide more information on the use of probiotics to restore workers' dysbiosis, we are aware that this is a very specific and novel topic and that the available literature is limited. We could include general information about probiotic potential to restore dysbiosis, but perhaps we believe it would not be in the topic covered in this Review and would make the review excessively long.  The available papers on this topic are listed in the review. We hope that more information on the subject will be available in the near future.

Reviewer 2 Report

This review article is very interesting. However, further revision is necessary before publication with reference to the following points.

(1)   This review deals with a wide range of subject area. What is the central question addressed by this review?

(2)   Recently, many similar reviews have been published and discussed several questions relating to the issues in this review. What insight does this review add to the subject area?

(3)   What is the novelty or originality of this article. A variety of recent review articles have addressed the relationships of gut microbiota with sleep disturbance, circadian rhythm, mental diseases, stress, metabolic diseases, etc. Author should clearly explain the difference between this review and these previous reviews.

(4)   The issues discussed in this review are now very hot and rapidly developing area. Very recently, more than several hundreds of similar papers and reviews relating to this manuscript have been published during 2022 to 2023. The authors should check the following recent reviews, and update the contents in this manuscript; 

1.     The microbiota-gut-brain axis in sleep disorders. Wang et al. Sleep Med Rev. 2022, 65:101691.

2.     The interplay between sleep and gut microbiota. Han et al. Brain Res Bull. 2022, 180:131-146.

3.     An important link between the gut microbiota and the circadian rhythm: imply for treatments of circadian rhythm sleep disorder. Tian et al. Food Sci Biotechnol. 2022, 31:155-164.

4.     The Oscillating Gut Microbiome and Its Effects on Host Circadian Biology. Litichevskiy et al. Annu Rev Nutr. 2022, 42:145-164.

5.     Sleep Deprivation and Gut Microbiota Dysbiosis: Current Understandings and Implications. Sun et al. Int J Mol Sci. 2023, 24:9603.

6.     Microbial circadian clocks: host-microbe interplay in diel cycles. Wollmuth et al. BMC Microbiol. 2023, 23:124.

7.     Circadian Disruption and Mental Health: The Chronotherapeutic Potential of Microbiome-Based and Dietary Strategies. Codoñer-Franch et al. Int J Mol Sci. 2023, 24:7579.

8.     Controlled light exposure and intermittent fasting as treatment strategies for metabolic syndrome and gut microbiome dysregulation in night shift workers. Bijnens et al. Physiol Behav. 2023, 263:114103.

9.     Microbial circadian clocks: host-microbe interplay in diel cycles. Wollmuth et al. BMC Microbiol. 2023, 23:124.

(5)   As the paragraph of L80-180 is too long, authors should revise it as compact as possible.

Author Response

Reviewer 2:

This article is very interesting and appropriate for publication. However, authors should further revise the following points. 

We wish to thank the reviewers for his revision of the manuscript. We expect our answers to be the adequate to take the paper into consideration and that it can be published in Nutrients.

(1) As the paragraph of L80-180 is too long, authors should revise it as compact as possible.

 Thank you for your comment, the paragraph has been amended.

(2) The issues in this review are now very hot and rapidly developing area. Very recently more than several hundreds of original papers and reviews relating to this manuscript have been published during 2022 to 2023. The authors should check the following recent reviews, and update the contents in this manuscript;  

  1. The microbiota-gut-brain axis in sleep disorders. Wang et al. Sleep Med Rev. 2022, 65:101691. 
  2. The interplay between sleep and gut microbiota. Han et al. Brain Res Bull. 2022, 180:131-146. 
  3. An important link between the gut microbiota and the circadian rhythm: imply for treatments of circadian rhythm sleep disorder. Tian et al. Food Sci Biotechnol. 2022, 31:155-164. 
  4. The Oscillating Gut Microbiome and Its Effects on Host Circadian Biology. Litichevskiy et al. Annu Rev Nutr. 2022, 42:145-164. 
  5. Sleep Deprivation and Gut Microbiota Dysbiosis: Current Understandings and Implications. Sun et al. Int J Mol Sci. 2023, 24:9603.
  6. Microbial circadian clocks: host-microbe interplay in diel cycles. Wollmuth et al. BMC Microbiol. 2023, 23:124.
  7. Circadian Disruption and Mental Health: The Chronotherapeutic Potential of Microbiome-Based and Dietary Strategies. Codoñer-Franch et al. Int J Mol Sci. 2023, 24:7579.
  8. Controlled light exposure and intermittent fasting as treatment strategies for metabolic syndrome and gut microbiome dysregulation in night shift workers. Bijnens et al. Physiol Behav. 2023, 263:114103. 
  9. Microbial circadian clocks: host-microbe interplay in diel cycles. Wollmuth et al. BMC Microbiol. 2023, 23:124.

Thank you for your comments. All the cited articles were included in the References section and discussed in the text along the main text of the manuscript.